# Cyclometalated and NNN Terpyridine Ruthenium Photocatalysts and Their Cytotoxic Activity

**DOI:** 10.3390/molecules29092146

**Published:** 2024-05-05

**Authors:** Maurizio Ballico, Dario Alessi, Eleonora Aneggi, Marta Busato, Daniele Zuccaccia, Lorenzo Allegri, Giuseppe Damante, Christian Jandl, Walter Baratta

**Affiliations:** 1Dipartimento di Scienze Agroalimentari, Ambientali e Animali, Università di Udine, Via Cotonificio 108, I-33100 Udine, Italy; alessi.dario@spes.uniud.it (D.A.); eleonora.aneggi@uniud.it (E.A.); marta.busato@uniud.it (M.B.); daniele.zuccaccia@uniud.it (D.Z.); 2Dipartimento di Medicina, Istituto di Genetica Medica, Università di Udine, Via Chiusaforte, F3, I-33100 Udine, Italy; allegri.lorenzo@spes.uniud.it (L.A.); giuseppe.damante@uniud.it (G.D.); 3Department of Chemistry & Catalysis Research Center, Technische Universität München, Ernst-Otto-Fischer-Str. 1, 85748 Garching bei München, Germany; christian.jandl@tum.de

**Keywords:** ruthenium, photocatalysis, transfer hydrogenation, cyclometalation, terpyridine, reduction, cytotoxicity

## Abstract

The cyclometalated terpyridine complexes [Ru(η^2^-OAc)(NC-tpy)(PP)] (PP = dppb **1**, (*R*,*R*)-Skewphos **4**, (*S*,*S*)-Skewphos **5**) are easily obtained from the acetate derivatives [Ru(η^2^-OAc)_2_(PP)] (PP = dppb, (*R*,*R*)-Skewphos **2**, (*S*,*S*)-Skewphos **3**) and tpy in methanol by elimination of AcOH. The precursors **2**, **3** are prepared from [Ru(η^2^-OAc)_2_(PPh_3_)_2_] and Skewphos in cyclohexane. Conversely, the NNN complexes [Ru(η^1^-OAc)(NNN-tpy)(PP)]OAc (PP = (*R*,*R*)-Skewphos **6**, (*S*,*S*)-Skewphos **7**) are synthesized in a one pot reaction from [Ru(η^2^-OAc)_2_(PPh_3_)_2_], PP and tpy in methanol. The neutral NC-tpy **1**, **4**, **5** and cationic NNN-tpy **6**, **7** complexes catalyze the transfer hydrogenation of acetophenone (S/C = 1000) in 2-propanol with NaO*i*Pr under light irradiation at 30 °C. Formation of (*S*)-1-phenylethanol has been observed with **4**, **6** in a MeOH/*i*PrOH mixture, whereas the *R*-enantiomer is obtained with **5**, **7** (50–52% *ee*). The tpy complexes show cytotoxic activity against the anaplastic thyroid cancer 8505C and SW1736 cell lines (ED_50_ = 0.31–8.53 *µ*M), with the cationic **7** displaying an ED_50_ of 0.31 *µ*M, four times lower compared to the enantiomer **6**.

## 1. Introduction

The design of efficient homogeneous catalysts for selective organic transformations occurring under benign conditions is an issue of great concern for the preparation of a number of value-added products [1]. In this context, the ruthenium catalysts [RuCl(η^6^-arene)(TsDPEN)] [2], [RuCl_2_(PP)(NN)] [3,4,5,6] (PP = diphosphine, NN = diamine, amino-pyridine), developed by Noyori, have found broad applications in the asymmetric hydrogenation with H_2_ [7] and transfer hydrogenation (TH) with 2-propanol of carbonyl compounds to alcohols, via bifunctional catalysis [8,9]. Conversely, the ruthenium complexes [Ru(bpy)_3_]X_2_ (X = Cl, PF_6_), containing bpy (2,2′-bipyridine) as a non-innocent ligand, have been used in C-X (X = C, N, O) coupling reactions in the presence of light via photochemical processes [10,11,12]. Notably, these complexes have been sparingly described in the photoreduction of carbonyl compounds, the Ru(bpy)_3_]^2+^/ viologen couple has been found to reduce 2-phenyl-2-oxoethanoic acid with triethanolamine (TEOA) as a sacrificial hydrogen donor [13]. In order to achieve efficient light-activated reactions, the choice of suitable stable ligand is crucial.

The tridentate tpy (2,2′:6′,2″-terpyridine) ligand has been used to prepare robust photocatalysts with good conjugation between the aromatic rings and the metal [14]. Tpy can also behave as a mono N or bidentate NN ligand [15,16], while the cyclometalated NC mode has been barely reported for Ir [17], Zn [18], Pd [19], and Pt [20,21] complexes, and no examples have been described for ruthenium. Regarding the tpy derivatives [22,23], [RuCl_n_(tpy)(PPh_3_)_3-n_]X_2-n_ (*n* = 1, 2) [24,25] and [RuCl_2_(*p*-cymene)]_2_/tpy [26] have proven to catalyze the reduction of carbonyl and aromatic nitro compounds, respectively, at a high temperature. Conversely, under irradiation, [RuCl_2_(tpy)(2,2′-bisquinoline)] catalyzes the TH of NAD^+^ to NADH with HCO_2_Na in water [27], [RuCl(tpy)(diphosphine)]Cl [28] catalyzes the TH of carbonyl compounds with 2-propanol, while the Ru(tpy)_2_^2+^ complexes generate hydrogen from TEOA [29,30]. Interestingly, in the electrochemical CO_2_ reduction, [Ru(tpy)(bis-carbene)(MeCN)][PF_6_]_2_ has proven to increase the rate 10-fold upon visible light illumination via a photon-assisted electrocatalysis [31,32].

In order to develop ruthenium derivatives that can find applications in catalysis and medicine, we have isolated a number of carboxylate derivatives [Ru(η^1^-OAc)_2_(PP)(en)] [33], [Ru(η^2^-OAc)(CO)(PP)(NN)]OAc [34,35], [Ru(η^1^-OAc)(CNN)(PP)] [36], which efficiently catalyze the reduction of carbonyl compounds, through a rapid Ru-OCOR carboxylate displacement. Interestingly, the complexes [Ru(η^1^-OAc)(CO)(PP)(phen)]OAc [37,38] showed high cytotoxicity against cancer cell lines and reacted with NADH as a hydrogen donor, affording Ru-H species [39] that may play a role in disturbing the cellular redox homeostasis [40,41,42]. It is worth noting that ruthenium carboxylates are reactive species which can be employed for the synthesis of electron reach ruthenium cyclometalated complexes via a concerted carboxylate-assisted deprotonation process [43,44,45,46,47], which can find applications in catalysis [48,49,50,51,52,53,54,55,56,57], photochemistry [58,59,60], and medicine [61,62]. 

Herein, we report a straightforward preparation of neutral cyclometalated [Ru(η^2^-OAc)(NC-tpy)(PP)] and cationic [Ru(η^1^-OAc)(NNN-tpy)(PP)]OAc (PP = diphosphine) terpyridine complexes starting from ruthenium acetate precursors. The derivatives containing a chiral diphosphine show asymmetric photocatalytic transfer hydrogenation of acetophenone and cytotoxic activity toward anaplastic thyroid cancer cell lines.

## 2. Results and Discussion

### 2.1. Synthesis of NC– and NNN–Terpyridine Ruthenium Complexes with Diphosphine Ligands

Treatment of the complex [Ru(η^2^-OAc)_2_(dppb)] with one equiv. of tpy in methanol at 55 °C for 2 h afforded the neutral NC–terpyridine derivative [Ru(η^2^-OAc)(NC-tpy)(dppb)] (**1**), as yellow precipitate isolated in 74% yield, via a “rollover” cyclometalation of tpy and elimination of acetic acid (Figure 1).

The ^31^P{^1^H} NMR spectrum of **1** in CD_2_Cl_2_ displays two doublets at δ 56.7 and 52.0 with a ^2^*J*(P,P) of 37.1 Hz, for the phosphorous trans to O and N atoms, respectively, as inferred from 2D ^1^H-^31^P HMBC NMR spectrum (Appendix A). The signals of the H6 and H6″ tpy protons are at δ_H_ 8.61 and 8.48, and the latter upfield shifted compared to the free ligand (δ 8.69) [62] (Figure 1). 

In the ^13^C{^1^H} NMR spectrum, the cyclometalated carbon C3′ appears at δ 182.7 (^2^*J*(C,P) = 18.0 and 8.4 Hz), whereas the signal at δ 184.5 is attributed to the carboxylate CO group. The resonances of the C6 and C6″ carbons are at δ 148.6 and 148.5, close to that of free tpy (δ 149.5) [63], whereas the C4′ carbon atom of the cyclometalated pyridine is significantly downfield shifted at δ 154.5 (Δδ = 16.3) and coupled with a phosphorous atom (^2^*J*(C,P) = 3.7 Hz). The structure of **1** in the solid state was confirmed by an X-ray diffraction experiment (Figure 2). 

Complex **1** crystallizes in a pseudo-octahedral geometry, showing a cyclometalated NC-terpyridine, a diphosphine and a chelate acetate ligand. The distortions arise from the small O1–Ru–O2 angle of the acetate (58.52(8)°), with similar Ru–O bond distances of 2.256(3) and 2.231(2) Å, not affected by the different *trans* P and C ligands. The Ru1–N1 (2.114(3) Å) and the Ru1–C7 (2.026(4) Å) lengths are in line with those of tpy [64,65,66,67], and NC-cyclometalated [65,68,69] ruthenium complexes. The X-ray analysis shows the presence of additional intramolecular π-π interactions between a phenyl group of dppb and the N-coordinated pyridine ring, in agreement with the behavior of **1** in solution with one phenyl displaying an upfield ^1^H NMR signal (δ_H_ 5.93). Although the “rollover” cyclometalation of tpy, affording a bidentate NC-ligand with a pendant pyridine, has been sparingly described for Pd, Pt and Zn complexes [16,18,20,21], no examples of this type of tpy coordination at ruthenium have been reported. It is worth noting that this ruthenium C-H activation may allow for the functionalization of tpy at the 3′ and 5′ positions of the internal pyridine [15].

Following the procedure described for **1**, chiral NC-terpyridine complexes have been obtained from diacetate ruthenium precursors containing chiral diphosphines. Thus, treatment of [Ru(η^2^-OAc)_2_(PPh_3_)_2_] with the (*R*,*R*)-Skewphos (1 equiv) in cyclohexane at reflux (4 h) results in the formation of the intermediate [Ru(η^2^-OAc)_2_((*R*,*R*)-Skewphos)] (**2**) isolated in 77% yield (Figure 2).

The ^31^P{^1^H} NMR spectrum of **2** in CD_3_OD shows a singlet at δ_P_ 65.9, whereas the ^1^H signal at δ_H_ 1.67 is for the two acetate methyl groups, in accordance with a complex of C_2_ symmetry. Similarly, the enantiomer [Ru(η^2^-OAc)_2_((*S*,*S*)-Skewphos)] (**3**) has been prepared from [Ru(η^2^-OAc)_2_(PPh_3_)_2_] and (*S*,*S*)-Skewphos and isolated in 83% yield (Figure 2).

Reaction of the precursor **2** with tpy (1 equiv) in methanol at 55 °C for 1 h results in the formation of the neutral NC-terpyridine derivative [Ru(η^2^-OAc)(NC-tpy)((*R*,*R*)-Skewphos)] (**4**), isolated in 65% yield as a single stereoisomer, as revealed by NMR analysis (Figure 2). The ^31^P{^1^H} NMR spectrum of **4** in CD_2_Cl_2_ shows two doublets at δ 70.6 and 54.0 with a ^2^*J*(P,P) value of 45.0 Hz for the phosphorous *trans* to the acetate O and N atoms, respectively (Appendix A). The resonances of the terminal H6 and H6″ pyridine protons of tpy are at δ_H_ 8.63 and 8.30, the latter showing a long-range coupling with the P atom at *δ*_P_ 54.0. Finally, the broad singlet at δ_C_ 184.1 is for the acetate CO and the doublet of doublets at δ_C_ 182.4 with ^2^*J*(C,P) = 16.1, and 8.8 Hz is for the cyclometalated Ru-*C*3′ atom. Also, in this case, the resonance of C4′ is significantly downfield shifted compared to that of the free ligand (Δδ = 15.9) [62]. According to the procedure described for **4**, the reaction of **3** with tpy affords the acetate [Ru(η^2^-OAc)(NC-tpy)((*S*,*S*)-Skewphos)] (**5**) isolated in 70% yield (Figure 2).

Conversely, cationic chiral NNN-terpyridine complexes have been obtained through a one-pot reaction starting from [Ru(η^2^-OAc)_2_(PPh_3_)_2_], PP and tpy, via the intermediate [Ru(OAc)_2_(PP)(PPh_3_)] (PP = Skewphos) in a protic solvent. Thus, treatment of [Ru(η^2^-OAc)_2_(PPh_3_)_2_] with one equivalent of (*R*,*R*)-Skewphos in MeOH at reflux for 4 h, followed by reaction with tpy, affords the derivative [Ru(η^1^-OAc)(NNN-tpy)((*R*,*R*)-Skewphos)]OAc (**6**), isolated as a single stereoisomer in 90% yield (Figure 3).

The ^31^P{^1^H} NMR spectrum of **6** in CD_3_OD shows two doublets at δ 52.8 and 36.1 with ^2^*J*(P,P) = 39.1 Hz for the phosphorous *trans* to O and N atoms, respectively, as inferred from the ^4^*J*(H,P) long-range coupling between the terminal *ortho* H6 and H6″ of tpy and the P *trans* to N, determined by a ^31^P-^1^H HMBC 2D NMR experiment (Appendix A). The ^1^H NMR spectrum displays the H6″ proton at δ 6.82, strongly upfield shifted (Δδ = 1.87) compared to the free ligand, with an NOE interaction with the *ortho* phenyl protons at δ 7.05 (Appendix A). Finally, the two resonances at δ_C_ 179.9 and 178.4 are for the bound and free acetate CO groups, respectively. Similarly, the enantiomer [Ru(η^1^-OAc)(NNN-tpy)((*S,S*)-Skewphos)]OAc (**7**) has been isolated in 86% yield from [Ru(η^1^-OAc)_2_(PPh_3_)_2_], (*S,S*)-Skewphos and tpy in methanol (Figure 3). Control ^31^P{^1^H} NMR experiments show that in methanol, [Ru(η^1^-OAc)_2_(PPh_3_)_2_] reacts with (*S*,*S*)-Skewphos at reflux, affording [Ru(η^1^-OAc)(η^2^-OAc)((*R*,*R*)-Skewphos))(PPh_3_)] as the main species, while **3** is present in a small amount (<3%) (Appendix A). 

The formation of the neutral and cationic tpy chiral ruthenium complexes is summarized in Figure 4.

Thus, [Ru(η^2^-OAc)_2_(PP)], obtained from [Ru(η^2^-OAc)_2_(PPh_3_)_2_] and PP in cyclohexane at reflux, reacts with tpy in methanol at 55 °C, affording the cyclometalated species [Ru(η^2^-OAc)(NC-tpy)(PP)] (PP = dppb, Skewphos). No cleavage of the Ru-C bond occurs by protonation with HOAc (3 equiv) in 2-propanol at 90 °C, whereas upon irradiation at 30 °C in methanol, the cationic derivative [Ru(η^1^-OAc)(NNN-tpy)(PP)]OAc is formed (52% of **6** from **5** in 12 h) (Appendix A). Conversely, these derivatives can be easily obtained by reaction of [Ru(η^2^-OAc)_2_(PPh_3_)_2_] with PP and tpy in methanol, by displacement of PPh_3_ and acetate (Figure 4). Thus, the facile metalation of the species [Ru(η^2^-OAc)_2_(PP)] with tpy, compared to [Ru(η^1^-OAc)(η^2^-OAc)(PP)(PPh_3_)], clearly indicates that the C-H cleavage, which requires a free coordination site, is prevented by the presence of a coordinated triphenylphosphine. It is worth noting that the acetate ligand plays a non-innocent role stabilizing coordinatively unsaturated intermediate species and acting as a weak base for the C-H bond activation, with the solvent (cyclohexane vs. methanol) strongly affecting the resulting products. 

### 2.2. TH of Acetophenone Photocatalyzed by Tpy Ruthenium Complexes

Complexes **1** and **4**–**7** (S/C = 1000) with NaO*i*Pr have been found to be active in the TH of acetophenone at 30 °C under light irradiation using a solar simulator (Figure 5), whereas **2**, **3**, which do not contain tpy, show no activity. The reactions were carried out using 2-propanol as the only hydrogen donor, without sacrificial agents (e.g., triethanolamine) and with no addition of photosensitizers. 

The cyclometalated **1** photocatalyzes the TH of acetophenone (0.1 M) in 2-propanol with NaO*i*Pr (2 mol %) at 30 °C, affording 93% conversion into 1-phenylethanol in 18 h and with TOF of 83 h^−1^ (entry 1 of Table 1), whereas in the dark, **1** is completely inactive, affording no significant formation of alcohol (<2 %) at reflux temperature.

With the chiral derivative **4**, acetophenone is quantitatively reduced in 2-propanol in 16 h to the alcohol racemate (TOF = 81 h^−1^, entry 2), whereas in an *i*PrOH/MeOH mixture (1/1 in volume), (*S*)-1-phenylethanol (93% conv.) is formed with 52% *ee* (TOF = 47 h^−1^, entry 3). Conversely, the enantiomer **5** gives (*R*)-1-phenylethanol (91% conv) with 50% *ee* in the *i*PrOH/MeOH mixture, whereas a racemic mixture is obtained in 2-propanol (entries 5 and 4). The cationic NNN–ruthenium complexes **6** and **7** afford 97 and 99% conversion of acetophenone in 9 h with TOF = 136 and 140 h^−1^, respectively (entries 6 and 8). By employment of the *i*PrOH/MeOH (1/1) mixture, **6** affords (*S*)-1-phenylethanol (92% conv) with 51% *ee* after 28 h of irradiation, while **7** gives the *R*-alcohol with 52% *ee* and 94% conv. (entries 7 and 9). An effect of the media on the catalytic asymmetric reduction of ketones with ruthenium catalysts has been described, resulting in some cases in an inversion of enantioselectivity by changing the polarity and bulkiness of the solvent [4,70]. It is worth noting that no reductive pinacol coupling of acetophenone has been observed upon irradiation in the presence of these tpy ruthenium complexes in basic 2-propanol [71].

Control experiments show that the neutral NC and cationic NNN complexes **5** and **7** are active only upon irradiation showing an “on/off” behavior and that the conversion follows a zero-order kinetic with respect to the substrate (Figure 3).

The comparison of the activity of the neutral **5** with the cationic **7** complexes, which show much the same *ee* values in the TH of acetophenone and a faster rate for **7** with respect to **5**, suggests that the catalysis occurs via similar NNN active species (Figure 6). NMR experiments show that **7** reacts with NaO*i*Pr (3 equiv) in 2-propanol-*d*^8^ at RT, under irradiation (30 min), affording the red-orange alkoxide [Ru(O*i*Pr)(NNN-tpy)((*S*,*S*)-Skewphos)](O*i*Pr) (**a**) species (δ_P_ 50.5 and 38.4 with ^2^*J*(P,P) = 35.2 Hz) [28] (Figure 6, Appendix A). Further irradiation (>2 h) leads to the brown mono hydride [RuH(NNN-tpy)((*S*,*S*)-Skewphos)](O*i*Pr) (**b**), as the main product, via a light-induced β-hydrogen elimination (Figure 6). The same hydride species **b** has been observed in the reaction of **5** with NaO*i*Pr (3 equiv) in 2-propanol/toluene-*d*^8^ upon irradiation (6 h), while in the dark, the hydride complex is not formed (Appendix A).

Based on these results, it is likely that with the NNN-tpy complexes, the photocatalytic TH occurs through the substitution of the coordinated acetate induced by light, affording the isopropoxide species **a**. Subsequently, the hydride **b** is formed via a light-driven β-*H*-elimination, which may occur through displacement of a pyridine moiety, with acetone extrusion [72]. The insertion of acetophenone into the Ru-H bond affords the alkoxide **c** that reacts with 2-propanol, leading to 1-phenylethanol and the isopropoxide **a** (Figure 6). Conversely, the use of the cyclometalated NC-tpy derivatives requires the conversion to NNN species. The asymmetric TH of acetophenone with the NC and NNN-tpy ruthenium complexes **4**–**7** indicates that this reduction takes place through a well-defined and robust chiral photocatalyst, without release of the N and P ligands.

### 2.3. Effects of Ruthenium Complexes on Cell Viability in ATC Cell Lines

Anaplastic thyroid cancer (ATC), while rare, remains one of the deadliest cancers known, showing a median overall survival of 3 months [73]. The lack of a standardized treatment protocol for the therapy of this type of neoplasm has resulted in a strong pressure to search for new therapeutic approaches in the cure of this cancer. Several therapeutic strategies were thus developed, ranging from more classical methods such as inhibition of cyclin-dependent kinases [74], to more innovative ones including the use of epigenetic drugs [75,76,77]. However, so far, all efforts made in the search for new molecules that can counteract the very high mortality of ATC have often been thwarted by the relative ease with which cancer cells are able to gain drug resistance. For these reasons, the development of new molecules that can increase ATC treatment options is crucial in an effort to extend the life expectancy. A preliminary assessment of the effects of the compounds under consideration involved studying their effectiveness in terms of cell viability. In order to evaluate the antitumor efficacy of ruthenium compounds, they were administered to ATC cells (SW1736 and 8505C) and to a non-tumorigenic thyroid cell line (Nthy-ori 3-1) at increasing doses, and an MTT assay was performed. Once the effects in terms of cell viability were observed, the effective dose 50 (ED_50_) was calculated by interpolation of the scatter plot curve (dose/effect). The two lines of ATC were similarly sensitive to each of the compounds tested, with the ED_50_ spanning from 0.3 to 8 µM, calculated at a 72 h time point (Table 2). 

Overall, all tested compounds proved less effective at reducing the cell viability of nontumorigenic cells. This is evidenced by the fact that ED_50_ in Nthy-ori 3-1 cells was consistently higher than that of SW1736 and 8505C, with increases ranging from 1.4- to 16-fold. Interestingly, compound **7** showed the highest difference between effects in ATC lines and nontumorigenic cells (Table 2). The neutral complexes **1**, **4** and **5** show moderate cytotoxicity, with the Skewphos derivatives being more efficient with respect to the dppb one, but no effect of chirality has been observed. For the cationic complexes, **7** bearing (*S*,*S*)-Skewphos displays a cytotoxicity (ED_50_ = 0.31 µM) four times higher with respect to its enantiomer **6** (ED_50_ = 1.39 µM). In addition, the related chloride [RuCl(NNN-tpy)((*S*,*S*)-Skewphos)]PF_6_ shows a higher ED_50_ value of 2.63 µM, indicating that the cell viability depends on the chirality of the complex and the nature of the anionic ligand, with the acetate derivative being more cytotoxic with respect to the chloride one. These chiral tpy acetate compounds show ED_50_ 2 to 20 times lower than cisplatin, confirming that these derivatives are more efficient than the classical chemotherapy agents in reducing cell viability in ATC cells. Viability effects were also observed on a non-tumor line, although they were significantly less relevant than in ATC cells. Analysis of cell viability alone is not sufficient to formulate hypotheses about the mechanism of action of these molecules, but it is presumable that they act at the level of the cell cycle or cell proliferation. For this reason, noncancer cells also experience their effects even if attenuated, since, as an in vitro model, they are immortalized and subject to a high rate of cell proliferation. The present data on ruthenium compounds on cell viability should be considered as the first step, as well as the starting point of further, more specific and more in-depth studies, aimed at evaluating other biological effects (cell aggressiveness, change in gene expression pattern) as well. Despite the preliminary nature of the results, the evidence of greater efficacy of these compounds than cisplatin is a very encouraging indication, especially considering that one of the main problems in the management of ATC is the high growth rate of this tumor, which makes blocking proliferation necessary as a first approach before enacting more targeted therapies. In addition, the lower ED_50_ of the compounds here investigated compared with cisplatin could suggest the use at lower doses, thus limiting the known adverse effects.

## 3. Materials and Methods

### 3.1. General Experimental Information

All reactions were carried out under an argon atmosphere using standard Schlenk techniques. The solvents were carefully dried by standard methods and distilled under argon before use. The ruthenium complexes [RuCl_2_(PPh_3_)_3_] [78], [RuCl_2_(dppb)(PPh_3_)] [79] and [Ru(η^2^-OAc)_2_(dppb)] [80] were prepared according to the literature procedures, whereas all other chemicals were purchased from Merck and Strem and used without further purification. NMR measurements were recorded on an Avance III HD NMR 400 spectrometer. Chemical shifts (ppm) are relative to TMS for ^1^H and ^13^C{^1^H}, whereas H_3_PO_4_ was used for ^31^P{^1^H}. The atom-numbering scheme for the NMR assignment of the terpyridine ligand in the ruthenium complexes is presented in Figure 1. Elemental analyses (C, H, N) were carried out with a Carlo Erba 1106 analyzer, whereas GC analyses were performed with a Varian CP-3380 gas chromatograph equipped with a 25 m length MEGADEX-ETTBDMS-β chiral column, with hydrogen (5 psi) as the carrier gas and flame ionization detector (FID). The injector and detector temperature was 250 °C, with initial T = 95 °C ramped to 140 °C at 3 °C/min for a total of 20 min of analysis. The t_R_ of acetophenone was 7.55 min, while the t_R_ of (*R*)- and (*S*)-1-phenylethanol was 10.49 min and 10.71 min, respectively.

### 3.2. Experimental Synthetic Procedure and Characterization Data for Ruthenium Complexes

Synthesis of [Ru(η^2^-OAc)(NC-tpy)(dppb)] (**1**).

[Ru(η^2^-OAc)(dppb)] (100.0 mg, 0.155 mmol) and tpy (37.0 mg, 0.159 mmol, 1.02 equiv) were dissolved in methanol (5 mL) and stirred at 55 °C for 2 h until a yellow precipitate was formed. The solid was filtered, washed with methanol (1 mL) and *n*-pentane (5 × 5 mL) and dried under reduced pressure. Yield: 93.9 mg (74%). Elemental analysis calcd (%) for C_45_H_41_N_3_O_2_P_2_Ru (818.86): C 66.01, H 5.05, N 5.13; found: C 65.95, H, 5.10, N 5.20. ^1^H NMR (400.1 MHz, CD_2_Cl_2_, 25 °C): δ 8.61 (dd, ^3^*J*(H,H) = 5.0 Hz, ^4^*J*(H,H) = 1.8 Hz, 1H; tpy (H6″)), 8.52 (d, ^3^*J*(H,H) = 8.0 Hz, 1H; tpy (H3″)), 8.48 (br d, ^3^*J*(H,H) = 5.8 Hz, 1H; tpy (H6)), 8.11 (br t, ^3^*J*(H,H) = 7.4 Hz, 2H; Ph), 7.94 (d, ^3^*J*(H,H) = 8.0 Hz, 1H; tpy (H3)), 7.85 (t, ^3^*J*(H,H) = 8.2 Hz, 2H; Ph), 7.81 (td, ^3^*J*(H,H) = 7.6 Hz, ^4^*J*(H,H) = 1.7 Hz, 1H; tpy (H4″)), 7.63–7.57 (m, 3H; Ph), 7.56–7.43 (m, 4H; Ph and tpy (H5′), (H4), (H4′)), 7.38 (td, ^3^*J*(H,H) = 7.4 Hz, ^4^*J*(H,H) = 1.8 Hz, 2H; Ph), 7.32–7.27 (m, 3H; Ph), 7.26–7.19 (m, 3H; Ph and tpy (H5″)), 6.96 (t, ^3^*J*(H,H) = 6.1 Hz, 1H; tpy (H5)), 6.79 (td, ^3^*J*(H,H) = 7.5 Hz, ^4^*J*(H,H) = 1.3 Hz, 1H; Ph), 6.56 (td, ^3^*J*(H,H) = 7.8 Hz, ^4^*J*(H,H) = 2.1 Hz, 2H; Ph), 5.93 (t, ^3^*J*(H,H) = 8.4 Hz, 2H; Ph), 3.03 (*pseudo*-q, *J* = 13.0 Hz, 1H; PCH_2_), 2.56 (tt, *J*(H,P) = 13.6 Hz, *J*(H,H) = 3.1 Hz, 1H; PCH_2_), 2.41–1.78 (m, 4H; PCH_2_CH_2_), 1.65–1.46 (m, 1H; CH_2_), 1.22 (s, 3H; CH_3_CO). ^13^C{^1^H} NMR (100.6 MHz, CD_2_Cl_2_, 25 °C): δ 184.5 (s; *C*OCH_3_), 182.7 (dd, ^2^*J*(C,P) = 18.0 Hz, ^2^*J*(C,P) = 8.4 Hz; tpy (C3′)-Ru), 163.7 (s; *ipso* tpy (C2)), 163.0 (s; *ipso* tpy (C2′)), 158.5 (s; *ipso* tpy (C2″)), 154.5 (d, ^3^*J*(C,P) = 3.7 Hz; tpy (C4′)), 148.6 (br s; tpy (C6″)), 148.5 (br s; tpy (C6)), 147.0 (s; *ipso* tpy (C6′)), 140.6 (d, ^1^*J*(C,P) = 34.0 Hz; *ipso*-Ph), 139.4 (d, ^1^*J*(C,P) = 43.7 Hz; *ipso*-Ph), 136.4 (s; tpy (C4″), 135.5 (s; tpy (C4), 134.6-126.4 (m; Ph), 122.1 (d, ^4^*J*(C,P) = 2.3 Hz; tpy (C5), 122.0 (s; tpy (C5″), 119.9 (s; tpy (C3)), 119.4 (s; tpy (C3″)), 117.2 (s; tpy (C5′), 30.6 (d, ^1^*J*(C,P) = 25.4 Hz; P*C*H_2_), 27.5 (d, ^1^*J*(C,P) = 30.7 Hz; P*C*H_2_), 25.7 (br s; *C*H_2_), 24.1 (s; OCO*C*H_3_), 22.2 (br s; *C*H_2_). ^31^P{^1^H} NMR (162.0 MHz, CD_2_Cl_2_, 25 °C): δ 56.7 (d, ^2^*J*(P,P) = 37.1 Hz), 52.0 (d, ^2^*J*(P,P) = 37.1 Hz).

Synthesis of [Ru(η^2^-OAc)_2_((R,R)-Skewphos)] (**2**).

[Ru(η^2^-OAc)(PPh_3_)_2_] (200.0 mg, 0.268 mmol) and (*R*,*R*)-Skewphos (120.8 mg, 0.274 mmol, 1.02 equiv) were suspended in cyclohexane (10 mL) and stirred at reflux for 4 h until a yellow solution was formed. The solvent was removed under reduced pressure, and *n*-heptane (10 mL) was added to the residue. The suspension was stirred at room temperature for 1 h then kept at −20 °C until a dark yellow precipitate was formed. The solid was filtered, washed with diethyl ether (2 × 2 mL) and *n*-heptane (3 × 5 mL) and dried under reduced pressure. The compound is air-sensitive and must be stored under inert gas. Yield: 136.6 mg (77%). Elemental analysis calcd (%) for C_33_H_36_O_4_P_2_Ru (659.66): C 60.09, H 5.50; found: C 60.15, H 5.45. ^1^H NMR (400.1 MHz, CD_3_OD, 25 °C): δ 7.99–6.94 (m, 20H; aromatic protons), 3.06–2.94 (m, 2H; PC*H*CH_3_), 2.04 (tt, ^3^*J*(H,H) = 20.6 Hz, ^3^*J*(H,P) = 5.2 Hz, 2H; CHC*H*_2_), 1.67 (s, 6H; OCOCH_3_), 0.95 (dd, ^3^*J*(H,P) = 13.3 Hz, ^3^*J*(H,H) = 6.9 Hz, 6H; PCHC*H*_3_). ^13^C{^1^H} NMR (100.6 MHz, CD_3_OD, 25 °C): δ 186.0 (br s; RuO*C*OCH_3_), 137.2 (d, ^1^*J*(C,P) = 10.3 Hz; *ipso*-Ph), 135.1–126.7 (m; aromatic carbon atoms), 36.4 (t, ^2^*J*(C,P) = 4.8 Hz; CH*C*H_2_), 26.3 (*pseudo*-t, *J*(C,P) = 16.5 Hz; P*C*HCH_3_), 23.3 (s; OCO*C*H_3_), 16.3 (br s; PCH*C*H_3_). ^31^P{^1^H} NMR (162.0 MHz, CD_3_OD, 25 °C): δ 65.9 (s).

Synthesis of [Ru(η^2^-OAc)_2_((*S*,*S*)-Skewphos)] (**3**).

Complex **3** was prepared following the procedure used for **2** employing (*S*,*S*)-Skewphos (120.8 mg, 0.274 mmol, 1.02 equiv) in place of (*R*,*R*)-Skewphos. Yield: 148.0 mg (83%). Elemental analysis calcd (%) for C_33_H_36_O_4_P_2_Ru (659.66): C 60.09, H 5.50; found: C 60.01, H 5.54. NMR data of **3** were identical to those of the enantiomer **2**.

Synthesis of [Ru(η^2^-OAc)(NC-tpy)((*R*,*R*)-Skewphos)] (**4**).

[Ru(η^2^-OAc)_2_((*R*,*R*)-Skewphos)] (**2**) (100.0 mg, 0.152 mmol) and tpy (36.5 mg, 0.156 mmol, 1.03 equiv) were dissolved in methanol (4 mL) and stirred at 55 °C for 1 h until a dark red solution was formed. The solvent was removed under reduced pressure and the residue was dissolved in diethyl ether (2 mL). The mixture was filtered to eliminate **6** that formed as a red product in a small amount. The orange solution was concentrated under reduced pressure to almost 0.5 mL, and *n*-heptane (5 mL) was added. The suspension was kept at −20 °C until a yellow precipitate was formed. The solid was filtered, washed with *n*-heptane (2 × 2 mL) and *n*-pentane (2 × 3 mL) and dried under reduced pressure. Yield: 82.0 mg (65%). Elemental analysis calcd (%) for C_46_H_43_N_3_O_2_P_2_Ru (832.89): C 66.34, H 5.20, N 5.05; found: C 66.25, H 5.15, N 4.96. ^1^H NMR (400.1 MHz, CD_2_Cl_2_, 25 °C): δ 8.63 (d, ^3^*J*(H,H) = 4.2 Hz, 1H; tpy (H6″)), 8.45 (d, ^3^*J*(H,H) = 8.0 Hz, 1H,; tpy (H3″)), 8.30 (d, ^3^*J*(H,H) = 5.4 Hz, 1H; tpy (H6)), 8.10 (d, ^3^*J*(H,H) = 8.1 Hz, 1H; tpy (H4′)), 7.92 (d, ^3^*J*(H,H) = 8.1 Hz, 1H; tpy (H3)), 7.82 (td, ^3^*J*(H,H) = 7.8 Hz, *J*(H,H) = 1.6 Hz, 1H; tpy (H4″)), 7.73–7.65 (m, 2H; Ph), 7.70 (d, ^3^*J*(H,H) = 8.1 Hz, 1H; tpy (H5′)), 7.62–7.47 (m, 3H; Ph and tpy (H4)), 7.45–7.29 (m, 12H; Ph), 7.28–7.23 (m, 1H; tpy (H5″)), 7.03 (t, ^3^*J*(H,H) = 6.2 Hz, 1H; tpy (H5)), 6.94 (t, ^3^*J*(H,H) = 7.2 Hz, 1H; Ph), 6.58 (td, ^3^*J*(H,H) = 7.9 Hz, ^4^*J*(H,H) = 1.5 Hz, 2H; Ph), 6.18 (t, ^3^*J*(H,H) = 8.2 Hz, 2H; Ph), 3.52–3.39 (m, 1H; PC*H*CH_3_), 2.58–2.46 (m, 1H; PC*H*CH_3_), 2.43–2.27 (m, 1H; CHC*H*_2_), 1.89 (dddd, ^2^*J*(H,H) = 32.4 Hz, ^3^*J*(H,P) = 29.0 Hz, ^3^*J*(H,H) = 14.4 Hz, ^3^*J*(H,P) = 3.0 Hz, 1H; CHC*H*_2_), 1.55 (s, 3H; OCOCH_3_), 1.52 (dd, ^3^*J*(H,H) = 12.5 Hz, ^3^*J*(H,H) = 7.5 Hz, 3H; CHC*H*_3_), 0.85 (dd, ^3^*J*(H,H) = 11.2 Hz, ^3^*J*(H,H) = 7.0 Hz, 3H; CHC*H*_3_). ^13^C{^1^H} NMR (100.6 MHz, CD_2_Cl_2_, 25 °C): δ 184.1 (s; RuO*C*OCH_3_), 182.4 (dd, ^2^*J*(C,P) = 16.1 Hz, ^2^*J*(C,P) = 8.8 Hz; tpy (C3′)-Ru), 163.6 (s; *ipso* tpy (C2)), 162.6 (s; *ipso* tpy (C2′)), 158.5 (s; *ipso* tpy (C2″)), 154.1 (d, ^3^*J*(C,P) = 3.7 Hz; tpy (C4′)), 148.8 (br s; tpy (C6″)), 148.6 (br s; tpy (C6)), 146.9 (s; *ipso* tpy (C6′)), 143.9 (d, ^1^*J*(C,P) = 35.2 Hz; *ipso*-Ph), 136.4 (s; tpy (C4″), 135.7 (s; tpy (C4), 134.9–126.4 (m; Ph), 122.6 (d, ^4^*J*(C,P) = 2.3 Hz; tpy (C5), 121.9 (s; tpy (C5″), 120.3 (s; tpy(C3)), 119.3 (s; tpy (C3″)), 117.7 (s; tpy(C5′), 38.5 (t, ^2^*J*(C,P) = 6.2 Hz; CH*C*H_2_), 33.3 (d, ^1^*J*(C,P) = 24.9 Hz; P*C*HCH_3_), 25.1 (s; OCO*C*H_3_), 20.6 (dd, ^1^*J*(C,P) = 31.5 Hz, ^3^*J*(C,P) = 5.1 Hz; P*C*HCH_3_), 19.3 (d, ^2^*J*(C,P) = 6.6 Hz; PCH*C*H_3_), 18.0 (br s; PCH*C*H_3_). ^31^P{^1^H} NMR (162.0 MHz, CD_2_Cl_2_, 25 °C): δ 70.6 (d, ^2^*J*(P,P) = 45.0 Hz), 54.0 (d, ^2^*J*(P,P) = 45.0 Hz).

Synthesis of [Ru(η^2^-OAc)(NC-tpy)((*S*,*S*)-Skewphos)] (**5**).

Complex **5** was prepared following the procedure used for **4,** employing [Ru(η^2^-OAc)_2_((*S*,*S*)-Skewphos)] (**3**) (100.0 mg, 0.152 mmol) in place of **2**. Yield: 88.0 mg (70%). Elemental analysis calcd (%) for C_46_H_43_N_3_O_2_P_2_Ru (832.89): C 66.34, H 5.20, N 5.05; found: C 66.27, H 5.18, N 5.06. NMR data of **5** were identical to those of the enantiomer **4**.

Synthesis of [Ru(η^1^-OAc)(NNN-tpy)((*R*,*R*)-Skewphos)]OAc (**6**).

[Ru(η^2^-OAc)_2_(PPh_3_)_3_] (100.0 mg, 0.134 mmol) and (*R*,*R*)-Skewphos (60.4 mg, 0.137 mmol, 1.02 equiv) were suspended in methanol (5 mL) and stirred at reflux for 4 h. The dark yellow solution was cooled at RT and concentrate at almost 2 mL under reduced pressure. Tpy (32.0 mg, 0.137 mmol, 1.02 equiv) was added, and the mixture was heated at reflux for 1 h until a dark red solution was formed. The addition of diethyl ether (10 mL) afforded the precipitation of the complex as a red-orange solid that was filtered, washed with of diethyl ether (5 × 10 mL), *n*-pentane (2 × 10 mL) and dried under reduced pressure. Yield: 108 mg (90%). Elemental analysis calcd (%) for C_48_H_47_N_3_O_4_P_2_Ru (892.94): C 64.57, H 5.31, N 4.71; found: C 64.55, H 5.25, N 4.66. ^1^H NMR (400.1 MHz, CD_3_OD, 25 °C): δ 8.86 (d, ^3^*J*(H,H) = 5.6 Hz, 1H; tpy (H6)), 8.29 (d, ^3^*J*(H,H) = 7.9 Hz, 1H; tpy (H3′)), 8.10–8.02 (m, 2H; tpy (H3″) and (H4′)) 8.01-7.91 (m, 3H; tpy (H3), (H4), (H5′)), 7.82 (br t, ^3^*J*(H,H) = 7.8 Hz, 1H; Ph), 7.74-7.65 (m, 3H; Ph and (H4″)), 7.48 (td, ^3^*J*(H,H) = 7.6 Hz, ^4^*J*(H,H) = 1.4 Hz, 1H; Ph), 7.43-7.24 (m, 7H; Ph and tpy (H5)), 7.15 (m, 2H; Ph), 7.09 (td, ^3^*J*(H,H) = 7.9 Hz, ^4^*J*(H,H) = 2.2 Hz, 2H; Ph), 7.05 (t, ^3^*J*(H,H) = 8.1 Hz, 2H; Ph), 6.93 (td, ^3^*J*(H,H) = 8.0 Hz, ^4^*J*(H,H) = 2.2 Hz, 2H; Ph), 6.87 (ddd, ^3^*J*(H,H) = 7.3 Hz, ^4^*J*(H,H) = 5.9 Hz, ^5^*J*(H,H) = 1.1 Hz, 1H; tpy (H5″)), 6.82 (d, ^3^*J*(H,H) = 5.6 Hz, 1H; tpy (H6″)), 6.39 (t, ^3^*J*(H,H) = 8.1 Hz, 2H; Ph), 3.99-3.85 (m, 1H; PC*H*CH_3_), 2.81 (qt, ^2^*J*(H,H) = 15.2 Hz, ^3^*J*(H,H) = 3.9 Hz; 1H; CHC*H*_2_), 2.68-2.56 (m, 1H; PC*H*CH_3_), 2.31-2.05 (m, 1H; CHC*H*_2_), 1.92 (s, 3H; OCOCH_3_), 1.55 (dd, ^3^*J*(H,P) = 12.4 Hz, ^3^*J*(H,H) = 7.6 Hz, 3H; CHC*H*_3_), 1.30 (s, 3H; RuOCOCH_3_), 0.60 (dd, ^3^*J*(H,P) = 12.0 Hz, ^3^*J*(H,H) = 6.8 Hz, 3H; CHC*H*_3_). ^13^C{^1^H} NMR (100.6 MHz, CD_3_OD, 25 °C): δ 179.9 (s; RuO*C*OCH_3_), 178.4 (s; O*C*OCH_3_), 160.5 (d, ^3^*J*(C,P) = 2.2 Hz; *ipso* tpy (C2)), 159.2 (d, ^3^*J*(C,P) = 2.9 Hz; tpy (C6)), 158.6 (d, ^3^*J*(C,P) = 2.9 Hz; *ipso* tpy (C2′)), 157.8 (s; *ipso* tpy (C2″), 155.5 (s; tpy (C6″)), 155.4 (s; *ipso* tpy (C6′)), 141.6 (d, ^1^*J*(C,P) = 35.9 Hz; ipso Ph), 138.0 (s; tpy (C4″)), 137.5 (s; tpy (C4)), 137.2 (s; tpy (C4′)), 136.4-126.8 (m; Ph), 126.0 (s; tpy (C5)), 125.3 (s; tpy (C5″)), 123.5 (s; tpy (C5′)), 122.3 (s; tpy (C3″)), 122.2 (s; tpy (C3)), 121.5 (s; tpy (C3′), 37.0 (t, ^2^*J*(C,P) = 5.9 Hz; CH*C*H_2_), 32.4 (d, ^1^*J*(C,P) = 22.7 Hz; P*C*HCH_3_), 23.8 (d, ^4^*J*(C,P) = 3.7 Hz; RuOCO*C*H_3_), 22.5 (s; OCO*C*H_3_), 20.1 (dd, ^1^*J*(C,P) = 28.2 Hz, ^3^*J*(C,P) = 4.8 Hz; P*C*HCH_3_), 18.1 (d, ^2^*J*(C,P) = 5.9 Hz; PCH*C*H_3_), 16.9 (br s; PCH*C*H_3_). ^31^P{^1^H} NMR (162.0 MHz, CD_3_OD, 25 °C): δ 52.8 (d, ^2^*J*(P,P) = 39.1 Hz), 36.1 (d, ^2^*J*(P,P) = 39.1 Hz).

Synthesis of [Ru(η^1^-OAc)(NNN-tpy)((*S*,*S*)-Skewphos)]OAc (**7**).

Complex **7** was prepared following the procedure used for **6** employing (*S*,*S*)-Skewphos (60.4 mg, 0.137 mmol, 1.02 equiv) in place of (*R*,*R*)-Skewphos. Yield: 103.0 mg (86%). Elemental analysis calcd (%) for C_48_H_47_N_3_O_4_P_2_Ru (892.94): C 64.57, H 5.31, N 4.71; found: C 64.59, H 5.34, N 4.76. NMR data of **7** were identical to those of the enantiomer **6**.

### 3.3. Typical Procedure for the Photocatalytic TH of Acetophenone

The ruthenium catalyst solution used for the photocatalytic TH was prepared by dissolving the complexes **1**, **4–7** (0.02 mmol) in 2-propanol (5 mL). The catalyst solution (250 μL, 1.0 μmol) and a 0.1 M solution of NaO*i*Pr (200 μL, 20 μmol) in 2-propanol were added subsequently to the acetophenone solution (1.0 mmol) in 2-propanol or a 2-propanol/MeOH (1:1 *v*/*v*) mixture (final volume 10 mL). The resulting solutions were stirred in a thermostated water bath at 30 °C. Irradiation of the samples was carried out using a 300 W Xenon Arc Lamp (LSB530A, LOT-Oriel, Darmstadt, Germany), emitting in the range 200–2500 nm (solar simulator). Samples were purged with Ar at least 15 min before irradiation. The reaction was sampled by removing an aliquot of the reaction mixture, which was quenched by the addition of diethyl ether (1:1 *v*/*v*), filtered over a short silica pad and submitted to GC analysis. The base addition was considered as the start time of the reaction. The S/C molar ratio was 1000/1, whereas the base concentration was 2 mol% with respect to the ketone substrate (0.1 M).

### 3.4. Cytotoxicity Assays

#### 3.4.1. Cell Lines

In this study, we used two different human anaplastic thyroid cancer cell lines (SW1736 and 8505C) and a non-tumorigenic thyroid cell line (Nthy-ori 3-1) that were grown as previously described [81]. All cell lines have been validated by short tandem repeat and tested for being mycoplasma-free. Cells were grown in RPMI 1640 medium (EuroClone, Milan, Italy) supplemented with 10% fetal bovine serum (Gibco Invitrogen, Milan, Italy), 2 mM L-glutamine (EuroClone, Milan, Italy), and 50 mg/mL gentamicin (Gibco Invitrogen, Milan, Italy). Cells were maintained in a humidified incubator (5% CO_2_, 37 °C).

#### 3.4.2. MTT Cell Viability Assay

In order to test cell viability, we applied the methylthiazolyldiphenyl-tetrazolium bromide (MTT) assay as previously described [82]. SW1736, 8505C and Nthy-ori 3-1cells (3000 cells/well) were plated onto 96-well plates in 200 μL medium/well and were allowed to attach to the plate for 24 h (t_0_). Plates were then treated either with DMSO or with each of the different compounds at different concentrations for 72 h. Then, 4 mg/mL MTT (Merck, Darmstadt, Germany) was added to the cell medium, and cells were cultivated for another 4 h in the incubator. The supernatant was removed, 100 μL/well of DMSO (Merck, Darmstadt, Germany) was added, and the absorbance at 570 nm was measured. All experiments were run sixfold and cell viability was expressed as a fold change compared to control. ED_50_ was calculated by interpolation of the scatter plot obtained by crossing each dose with its own observed effect.

### 3.5. X-ray Crystallography

Single crystals of the complex **1** were obtained by slow cooling of a concentrated solution of the species in CH_2_Cl_2_/heptane. X-ray diffraction data were collected on a Bruker D8 Venture single crystal x-ray diffractometer equipped with a CPAD detector (Bruker Photon II), an IMS microsource with MoK_α_ radiation (*λ* = 0.71073 Å) and a Helios optic using the APEX3 Version 2019-1.0 software package. For additional details about collection and refining of data, see the Supporting Information. CCDC 2302606 contains the supplementary crystallographic data for this paper. These data are provided free of charge by The Cambridge Crystallographic Data Centre.

## 4. Conclusions

In summary, we have reported a straightforward preparation of a rare example of NC-cyclometalated terpyridine complexes [Ru(η^2^-OAc)(NC-tpy)(PP)] (PP = dppb, Skewphos) from the acetate compounds [Ru(η^2^-OAc)_2_(PP)] and tpy, the chiral derivatives being isolated as single stereoisomers. Conversely, the cationic NNN-terpyridine derivatives [Ru(η^1^-OAc)(Skewphos)(NNN-tpy)]OAc are prepared from [Ru(η^2^-OAc)_2_(PPh_3_)_2_], Skewphos and tpy. The neutral NC-tpy and the cationic NNN-tpy complexes catalyze the transfer hydrogenation of acetophenone under light irradiation at 30 °C and with an enantioselectivity of 50-52% with the chiral phosphine and using an *i*PrOH/MeOH mixture. The tpy complexes have proven to be cytotoxic against the anaplastic thyroid cancer 8505C and SW1736 cell lines, with ED_50_ values ranging from 0.31 to 8.53 µM. The NNN-tpy derivative with (*S*,*S*)-Skewphos displays an ED_50_ = 0.31 µM, four times higher compared to its enantiomer. Further studies are ongoing to broaden the chemistry of chiral ruthenium complexes based on polypyridine and phosphine ligands for photocatalytic transformations and for their use as metallodrugs.

## Data Availability

Crystallographic data for compound **1** have been deposited with the Cambridge Crystallographic Data Centre as supplementary publication number CCDC 22302606.

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
