# Peer review of "Cyclometalated and NNN Terpyridine Ruthenium Photocatalysts and Their Cytotoxic Activity"

_molecules, 2024, doi:10.3390/molecules29092146_

Round 1
Reviewer 1 Report
Comments and Suggestions for Authors
1) There needs to be a careful proofread and revision of the introduction, a clearer logic storyline should be provided to provide readers with a better reading experience.
2) The background of photocatalysts might be strengthened; especially the advantages of these materials over similar reported photocatalysts should be added.
3) What is the role of Rh in photocatalysts? Further explanation is needed for the mechanism.
4) Authors are encouraged to add some discussion of work and their widely applications in the field of photochemistry, some related works in ACS Appl. Mater. Interfaces., 2021, 13, 12463−12471; Molecules 2023, 28, 4507
5) The title is not appealing, it should be revised.
6) The abstract part is not well written, it should be revised according to the novelty of this work
7) The experimental part needs specific references to ensure the reproducibility of this work.
Comments on the Quality of English Languagerevsion
Author Response
Comments and Suggestions for Authors
1) There needs to be a careful proofread and revision of the introduction, a clearer logic storyline should be provided to provide readers with a better reading experience.
The introduction has been changed according to the referee suggestions
2) The background of photocatalysts might be strengthened; especially the advantages of these materials over similar reported photocatalysts should be added.
While several ruthenium photocatalysts have been employed in C-X (X = C, N, O) forming reaction (e.g. [Ru(bipy)3]2+), very few examples of transfer hydrogenation ruthenium catalysts active under light have been reported. These parts has been better explained in the introduction and relevant reviews have been cited (notes 10, 11, and 14).
3) What is the role of Rh in photocatalysts? Further explanation is needed for the mechanism.
As described in the text and in Scheme 4, the light adsorbed by the tpy ruthenium isopropoxide complex induces the formation of Ru-hydride species, via a photo beta-H elimination, which is responsible of the reduction of acetophenone. The photoredox properties of Ru polypyridine complexes have bene extensively described in the cited reviews (see above).
4) Authors are encouraged to add some discussion of work and their widely applications in the field of photochemistry, some related works in ACS Appl. Mater. Interfaces., 2021, 13, 12463−12471; Molecules 2023, 28, 4507.
This a preliminary work on the transfer hydrogenation from 2-propanol to acetophenone, photocatalyzed by Ru tpy complexes. The applications of these systems for other photochemical transformations is beyond this study. The article Molecules 2023, 28, 4507 deals with the synthesis of thiophene polymers which lead to hydrogen production from ascorbic acid under light and it has been added to the text. Conversely, the article ACS Appl. Mater. Interfaces., 2021, 13, 12463 regards to the oxidation of alcohols on heterogeneous catalysts in air and therefore it has not been added since is not related to this paper.
5) The title is not appealing, it should be revised.
The tile has been shortened “Cyclometalated and NNN Terpyridine Ruthenium Photocatalysts and their Cytotoxic Activity”
6) The abstract part is not well written, it should be revised according to the novelty of this work.
The abstract has been rewritten.
7) The experimental part needs specific references to ensure the reproducibility of this work.
The references for the preparation of the ruthenium precursors are reported in “General Experimental Information“, while the references for the cell lines growth methods and MTT assays are in the “Cytotoxicity Assays” paragraphs of the Material and Methods section.
Reviewer 2 Report
Comments and Suggestions for Authors
This paper reports a new Ru-complex which is chiral and is a photocatalyst. There are several similar molecules reported in literature, however this molecule is new and was confirmed by SCXRD. Additionally, ED50 was calculated for a comparative study of cytotoxicity with cisplatin. The manuscript could be interesting for the reader of the journal. I suggest accept after major revision.
the authors need to clarify the motivation part of this manuscript.
The methodologies behind the cell lines studies are still inadequate. It has to be elaborately explained.
Establishment of this new complex on a point of efficiency over cisplatin is non-trivial and thus needs more studies. Thus such claims should be modified.
The studies of cytotoxicity isnot in-depth however promising. Therefore, it should be considered preliminary and not having strong conclusion. Necessary modifications must be made.
The addition of chirality part is quit non-coherent and inconclusive as the catalyst has not be studied under any polarized light. The efficacies of those are not having a big differences like in asymmetric synthesis.
The representation of hν is inappropriate in all places. why ν is a superscript?
is there a special reason?
Author Response
Comments and Suggestions for Authors
This paper reports a new Ru-complex, which is chiral and is a photocatalyst. There are several similar molecules reported in literature, however this molecule is new and was confirmed by SCXRD. Additionally, ED50 was calculated for a comparative study of cytotoxicity with cisplatin. The manuscript could be interesting for the reader of the journal. I suggest accept after major revision.
The authors need to clarify the motivation part of this manuscript.
The introduction has been modify to clarify the motivation of this work.
The methodologies behind the cell lines studies are still inadequate. It has to be elaborately explained.
Establishment of this new complex on a point of efficiency over cisplatin is non-trivial and thus needs more studies. Thus, such claims should be modified.
The studies of cytotoxicity is not in-depth however promising. Therefore, it should be considered preliminary and not having strong conclusion. Necessary modifications must be made.
In the Results section, the reason why using cell lines from anaplastic thyroid cancer has been better delineated. In addition, a comparison with a non-tumorigenic cell line has been included. As suggested by the referee, we have modified the conclusions drawn on the basis of the results obtained on cell viability, reiterating the preliminary nature of the data and the need for further and more in-depth studies. This was added after the description of Table 2, at the end of Results and Discussion. The statistical analysis of nontumorogenic thyroid cell (Nthy-ori 3-1) viability has been added in the Supporting Information.
The addition of chirality part is quit non-coherent and inconclusive, as the catalyst has not be studied under any polarized light. The efficacies of those are not having a big difference like in asymmetric synthesis.
The complexes have been obtained as single stereoisomers with the C2 chiral diphosphine. As reported in the text, the chirality at the metal center induces asymmetric catalytic reduction of acetophenone (55% ee) a process that occurs in the presence of light to promote the formation of the catalytically active Ru-H species
The representation of hν is inappropriate in all places. Why ν is a superscript?
Is there a special reason?
The ν appears as superscript only after the pdf conversion, while in the original Word file it is correctly written.
Reviewer 3 Report
Comments and Suggestions for Authors
All that is needed would be an in vitro study with non-cancerous thyroid cells as controls. This would make the manuscript complete in nature.
Author Response
Comments and Suggestions for Authors
All that is needed would be an in vitro study with non-cancerous thyroid cells as controls. This would make the manuscript complete in nature.
As requested by the reviewer, effects of ruthenium compounds on nontumorigenic thyroid cells (Nthy-ori 3-1) have been investigated. Results are included in the revised Table 2 (as ED50), delineated in the Results section and commented in the Discussion.
Round 2
Reviewer 2 Report
Comments and Suggestions for Authors
The revised manuscript has added the required inputs that I raised concern with. The conclusion is thus coherent with the results and discussions and methodologies. I recommend accept.